# Convalescent plasma use in the USA was inversely correlated with COVID-19 mortality

Arturo Casadevall[1]*, Quigly Dragotakes[1], Patrick W Johnson[2], Jonathon W Senefeld[3], Stephen A Klassen[3], R Scott Wright[4], Michael J Joyner[3†], Nigel Paneth[5†], Rickey E Carter[2†]

[1]Department of Molecular Microbiology and Immunology, Johns Hopkins School of Public Health, Baltimore, United States; [2]Department of Quantitative Health Sciences, Mayo Clinic, Jacksonville, United States; [3]Department of Anesthesiology and Perioperative Medicine, Mayo Clinic, Rochester, United States; [4]Department of Cardiology, Mayo Clinic, Rochester, United States; [5]Department of Epidemiology and Biostatistics and Department of Pediatrics and Human Development, College of Human Medicine, Michigan State University, East Lansing, United States

## Abstract

**Background:** The US Food and Drug Administration authorized COVID-19 convalescent plasma (CCP) therapy for hospitalized COVID-19 patients via the Expanded Access Program (EAP) and the Emergency Use Authorization (EUA), leading to use in about 500,000 patients during the first year of the pandemic for the USA.

**Methods:** We tracked the number of CCP units dispensed to hospitals by blood banking organizations and correlated that usage with hospital admission and mortality data.

**Results:** CCP usage per admission peaked in Fall 2020, with more than 40% of inpatients estimated to have received CCP between late September and early November 2020. However, after randomized controlled trials failed to show a reduction in mortality, CCP usage per admission declined steadily to a nadir of less than 10% in March 2021. We found a strong inverse correlation ($r = -0.52$, p=0.002) between CCP usage per hospital admission and deaths occurring 2 weeks after admission, and this finding was robust to examination of deaths taking place 1, 2, or 3 weeks after admission. Changes in the number of hospital admissions, SARS-CoV-2 variants, and age of patients could not explain these findings. The retreat from CCP usage might have resulted in as many as 29,000 excess deaths from mid-November 2020 to February 2021.

**Conclusions:** A strong inverse correlation between CCP use and mortality per admission in the USA provides population-level evidence consistent with the notion that CCP reduces mortality in COVID-19 and suggests that the recent decline in usage could have resulted in excess deaths.

**Funding:** There was no specific funding for this study. AC was supported in part by RO1 HL059842 and R01 AI1520789; MJJ was supported in part by 5R35HL139854. This project has been funded in whole or in part with Federal funds from the Department of Health and Human Services; Office of the Assistant Secretary for Preparedness and Response; Biomedical Advanced Research and Development Authority under Contract No. 75A50120C00096.

*For correspondence: acasade1@jhu.edu

†These authors contributed equally to this work

Competing interests: The authors declare that no competing interests exist.

## Introduction

In the Spring of 2020, the USA embarked on a historic and unprecedented deployment of plasma derived from patients who survived COVID-19 (COVID-19 convalescent plasma [CCP]) for treatment of the disease, and 1 year into this effort more than 500,000 individuals have been treated. This

development was fueled by the lack of effective alternate therapies, a plentiful supply of plasma from an efficient and high-capacity blood banking network, motivated donors, and strong community partners. US Food and Drug Administration (FDA) regulatory oversight was provided first by its Expanded Access Program (EAP) in partnership with the Mayo Clinic, with first transfusion on early April 2020 (*Senefeld et al., 2021*), and then by its Emergency Use Authorization (EUA) of August 23, 2020, both of which restricted CCP use to hospitalized patients (*Anonymous, 2021b*).

The demonstration by the Summer of 2020 that CCP was safe (*Joyner et al., 2020a*; *Joyner et al., 2020b*), that antibody concentration in plasma correlated with survival in people treated before mechanical ventilation (*Joyner et al., 2021*) along with initial suggestions of efficacy (*Li et al., 2020*; *Liu et al., 2020*; *Salazar et al., 2020*), boosted its use. Nonetheless, the use of CCP rose rapidly without the ideal evidence base of efficacy from randomized controlled clinical trials (RCTs), since early RCTs though generally trending positively were inconclusive, mostly due to small size or premature termination as the epidemic abated in the early surge regions (*Casadevall et al., 2020*). Later in the pandemic, several larger RCTs reported no mortality benefit (*Agarwal et al., 2020*; *Horby et al., 2021*; *Simonovich et al., 2021*), raising doubts as to CCP efficacy. However, these latter trials were undertaken in hospitalized patients treated late in the course of disease and some used plasma with variable antibody levels (*Casadevall et al., 2020*), and contrasted with a highly successful trial in elderly patients treated within 3 days of illness onset prior to hospitalization (*Libster et al., 2021*). Despite potential explanations for the negative studies, the results of these studies were sometimes accompanied by editorials that reinforced the message of futility with the *British Medical Journal* calling CCP 'ineffective' (*Pathak, 2020*), *Nature Biotechnology* reported that CCP fell 'flat' (*Sheridan, 2020*), and *JAMA* published a meta-analysis of RCT concluding that there was no evidence of benefit from CCP therapy (*Janiaud et al., 2021*). On February 17, 2021, the *Wall Street Journal* reported that Mount Sinai Hospital, which had been a leader in deploying CCP and reported early encouraging results (*Liu et al., 2020*), had stopped using plasma in patients with COVID-19, and the report specifically mentioned that the negative results from CCP RCTs drove this decision (*Marcus, 2021*).

On March 13, 2021, the *New York Times* reported that COVID-19 mortality remained high with nearly 1500 daily deaths despite a drop in the number of new infections since earlier in the year (*Leatherby, 2021*). Consistent with this report, an analysis of treatment outcomes found that they worsened for the last 2 months of 2020 (*Garg et al., 2021*). This finding was surprising in light of an apparent reduction in the mortality of hospitalized patients as the epidemic progressed, thought to be from improved management of the disease as clinical experience grew (*Ledford, 2020*). Analyzing weekly reports from the blood banking industry, we noted that plasma use was on the decline, based on the ratio of units dispensed to hospital relative to hospital admissions. The increase in mortality combined with the reduction in CCP use led us to hypothesize first, that the two phenomena were related, and second, that the decline in CCP use reflected reduced use following the disappointing trial findings. We therefore examined the use of CCP units as a function of time, assessing the relationship of CCP use to COVID-19 mortality, denominating both plasma units and deaths to hospital admissions.

## Materials and methods

### Convalescent plasma usage

CCP usage was inferred from the distribution of plasma units to hospitals in the USA from data obtained from Blood Centers of America, Inc (BCA, West Warwick, RI). Data fields included collections, distributions to hospitals, distributions to research, or other use. This file consolidated all the reports from regional blood bank reports and provided a total of collected units and units distributed to hospitals. The data file did not have information on whether a unit was actually transfused but BCA can infer usage from hospital re-ordering information and there has been a strong correlation between the total number of units shipped to a hospital and the units transfused by that hospital. Hence, the CCP units dispensed to hospitals represent a reasonable proxy value for the total number of units being transfused to patients. To validate this assumption, we compared the numbers of plasma units dispensed to those used by the EAP. There was a powerful and significant correlation between the weekly counts of units distributed in the USA and those used to treat patients

as part of the EAP between April 6 and August 23, 2020 (Spearman's rho = 0.953, p<0.001) (*Figure 1—figure supplement 1*). Units transfused in the EAP were reported by providers as part of the official case report forms and each transfusion could comprise one or two units. During both the EAP and EUA epochs of CCP collection and distribution included in this analysis, reimbursement for units collected was provided by the US government. Additionally, most blood products used in the USA including CCP are collected and distributed by a network of regional or national blood banking organizations. In this context, the vast majority (>95%) of all convalescent plasma (CP) administered was likely captured in this analysis (*Senefeld et al., 2021*).

## Admission and mortality data

For population-level data on COVID-19 admissions and mortality, we relied on publicly available databases. Specifically, we used information from the Our World in Data (OWID) (https://ourworldindata.org/coronavirus) database. Data used for this analysis were downloaded on March 18, 2021, and are available as *Supplementary file 3*. We confirmed these findings using Centers for Disease Control (CDC, Atlanta, GA) data on admissions and deaths https://covid.cdc.gov/covid-data-tracker/#new-hospital-admissions and https://covid.cdc.gov/covid-data-tracker/#trends_dailytrendsdeaths. CDC data were downloaded on March 31, 2021, and are available together with other data used in this analysis at: https://github.com/eapPlasma/plasma_hesitancy/tree/main/rawdata.

## Statistical analysis

Preliminary descriptive analyses were used to explore the associations of the ratio of number of CCP units dispersed to the number of hospitalizations (CCP utilization ratio) with the ratio of national deaths to national admissions, the latter being a reasonable proxy for the case fatality rate (CFR). No individual-level data were available to link the mortality events directly to the individuals hospitalized to permit a calculation of the true CFR. To address this limitation, the mortality counts reported by the CDC were shifted to better align the deaths with the admitted patients. Since the overwhelming majority of COVID-19 deaths occur in hospitals (*Chua et al., 2021*; *Chuzi et al., 2020*), since CCP is only authorized for use in hospitals, and since death generally occurs a few days to weeks after admission, mortality was adjusted for the time lag between admission and death. The median time between admission and death has been reported as 9 days in the USA (*Horwitz et al., 2021*) and 6.7 days in Belgium (*Faes et al., 2020*). For the analysis, which was based on weekly aggregated data, a 2-week shift was selected to align the mortality with the median and upper quartile estimates in these reports. The Pearson's correlation coefficient was used to describe the relationship of the CCP utilization ratio with CFR. To further define this relationship, a linear statistical model was used to regress the utilization ratio onto CFR. This statistical model was weighted by the number of hospitalizations per week. The fit of the model was examined using standard residual-based diagnostic plots and the fit was deemed acceptable using only a linear fit of the CCP utilization ratio.

Three in silico scenarios were created to summarize the effect of alterations to the CCP utilization ratio using the fitted model. In scenario 1, the effect of maintenance of plasma usage was considered. To define maintenance of use, a weighted average of the utilization ratio over the months of August through September 2020 was estimated. This value was then used to estimate the number of deaths that would be expected to occur throughout the study period (admissions starting August 3, 2020 to February 22, 2021). In scenario 2, a constant 50% CCP utilization ratio was set over the entire study period. The utilization rate was approximately the value observed in the early October 2020 period. A final scenario estimated the CFR that may have been observed had CCP not been used at all (i.e., the y-intercept from the model). Model contrasts were used to estimate the change in expected deaths among these scenarios in addition to a fourth condition – the actual number of events reported by the CDC. Values are summarized based on the observed number of hospitalized patients over the study period along with the same values indexed into expected mortality events per 1000 hospitalizations. Pointwise confidence bands for each scenario were obtained by multiplying the model-predicted CFR and its associated 95% confidence interval by the number of hospitalizations per week. Cumulative summations were used to describe the differences in expected deaths over the entire study period. This analysis was repeated independently a third time using a weighted average of utilization ratio of the months October-November on a separate database (OWID) to investigate stability between reporting bodies. As a final method for estimating the

overall effect of the changes in CCP utilization, the CDC data were grouped into two time periods representing relative use. The difference in the CFR was used to estimate the expected value for the changes in the expected number of deaths.

Statistical analyses were conducted using R version 3.6.2 (*R Development Core Team, 2013*); 95% confidence intervals and two-sided p-values were used to summarize association and test for significance at the alpha = 0.05 level of significance, respectively.

## Results

### CP use

The FDA first allowed compassionate use of CCP on a case-by-case basis in late March 2020, but very quickly initiated the EAP in early April 2020, which was followed by EUA on August 23, 2020, broadening its use. Distribution of CCP to hospitals rose to 25,000–30,000 weekly units by the December 2020 to January 2021 time period, but this rise in plasma distribution largely reflected the great increase in hospital admissions for COVID in those months (*Figure 1*). When CCP distributions are analyzed as a function of the number of new hospital admissions per week, peak utilization per capita occurred much earlier, in early October 2020 and declined sharply in the following months (*Figure 2*).

### Correlation between CCP and mortality

To explore whether there was a relationship between CCP distribution and mortality, we first compared the doses per patient versus reported COVID-19 deaths per hospital admission from publicly available databases (*Figure 2*). The comparison of curves showed a trough in deaths per admission coinciding with the peak of CCP usage per admission. A plot of mortality versus doses per

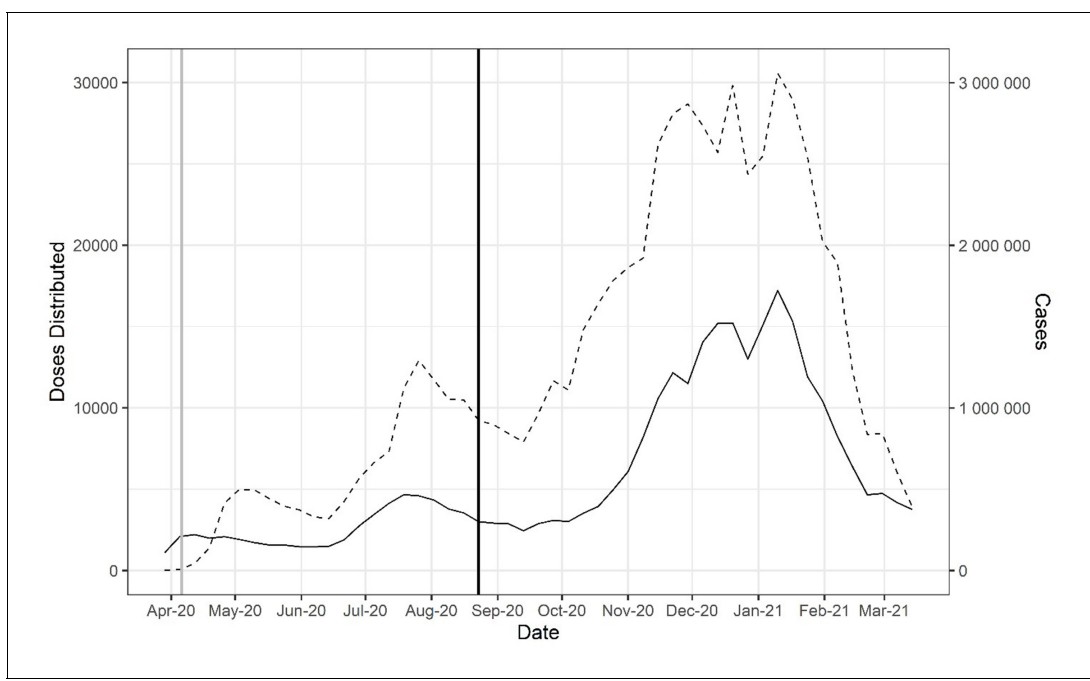

**Figure 1.** Doses of COVID-19 convalescent plasma (CCP) distributed in the USA by the American Red Cross and American Blood Centers (dashed) and total COVID-19 cases in the USA reported in Our World in Data (OWID) database (solid). The vertical black line marks August 23, 2020, when the US Food and Drug Administration (FDA) announced that Emergency Use Authorization for CCP in the USA. The vertical gray line marks April 4, 2020, as the start of the Emergency Access Program.

The online version of this article includes the following source data and figure supplement(s) for figure 1:

**Source data 1.** Source data for *Figure 1*.

**Figure supplement 1.** Correlation of convalescent plasma distribution and usage within the Expanded Access Program (EAP).

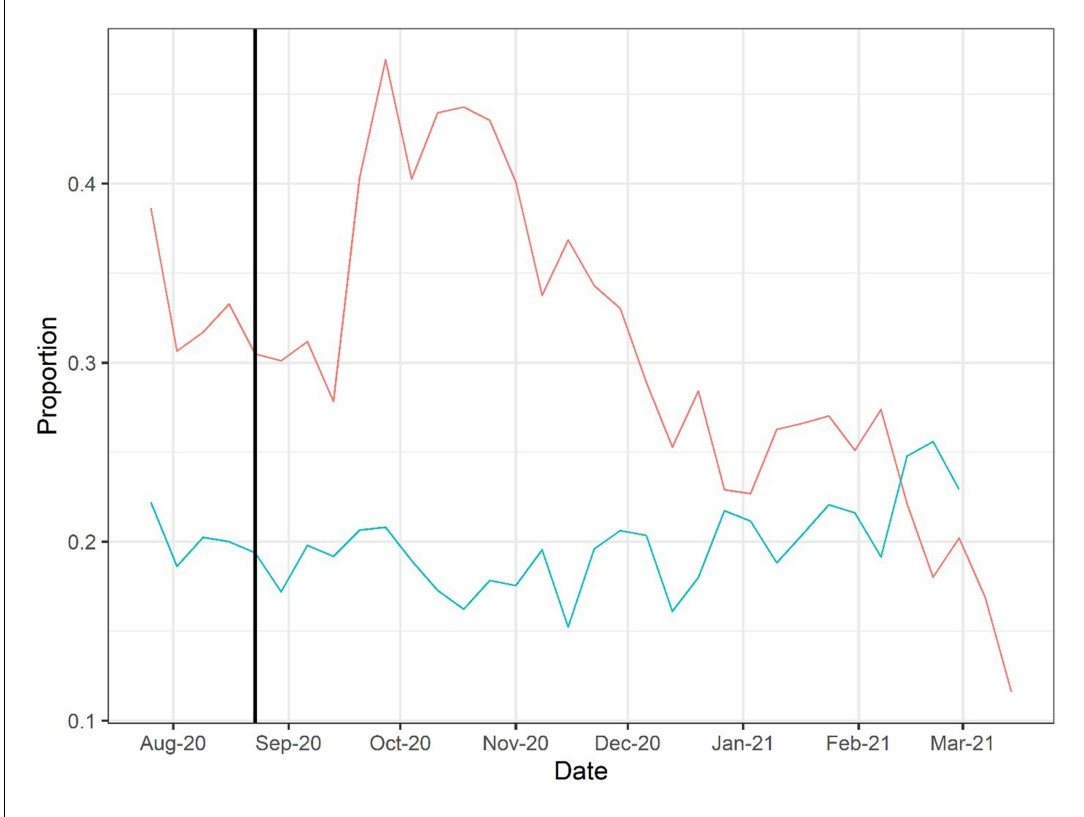

**Figure 2.** Doses of COVID-19 convalescent plasma (CCP) per hospital admission (red) and mortality calculated as deaths per hospital admission (green) using Our World in Data (OWID) database. To account for time between admission to death, deaths from 2 weeks after admission are used to calculate mortality. The vertical line marks August 23, 2020, when the US Food and Drug Administration (FDA) announced that Emergency Use Authorization for CCP in the USA.

The online version of this article includes the following source data for figure 2:

**Source data 1.** Source data for *Figure 2*.

hospitalized patient using mortality per admission data from the OWID database revealed a strong negative correlation (Pearson's correlation coefficient of −0.52 with p=0.002) (*Figure 3*). Similar results were obtained with the CDC database. To account for lags in the reporting of death that vary by state, we also investigated whether this correlation was maintained while adding weeks to the time between admission and death (*Figure 3—figure supplement 1*). Additionally, if plasma use is divided into quintiles from lowest using weeks to highest using weeks, a graded relationship between use of plasma and mortality is apparent (*Figure 3—figure supplement 2*). To investigate whether a change in the age demographics of the hospitalized population could account for the observed inverse correlation, we compared the shifted mortality is compared to the percent of hospitalized patients 65+ each week as reported by the CDC and found no significant correlation between these two variables, suggesting changes in mortality are not explainable by an increase in hospitalized high-risk patients (*Figure 3—figure supplement 3*).

## Estimates of excess deaths

The linear model using the CCP utilization ratio to predict the CFR fits the data well in that this model explained 25% ($R^2$ = 0.254) of the variance of CFR using only the CCP utilization ratio. The model estimated that the CFR decreased by 1.8 percentage points for every 10 percentage point increase in the rate of utilization of CCP (p=0.004). The linear regression analysis yielded a mortality in patients not receiving CCP of 25.2% as the y-intercept. A comparison of this number with that from US studies shows reasonable agreement between with the average mortality of 23.5% in patients not receiving CCP (*Supplementary file 1*). This percentage also closely matches the 24%

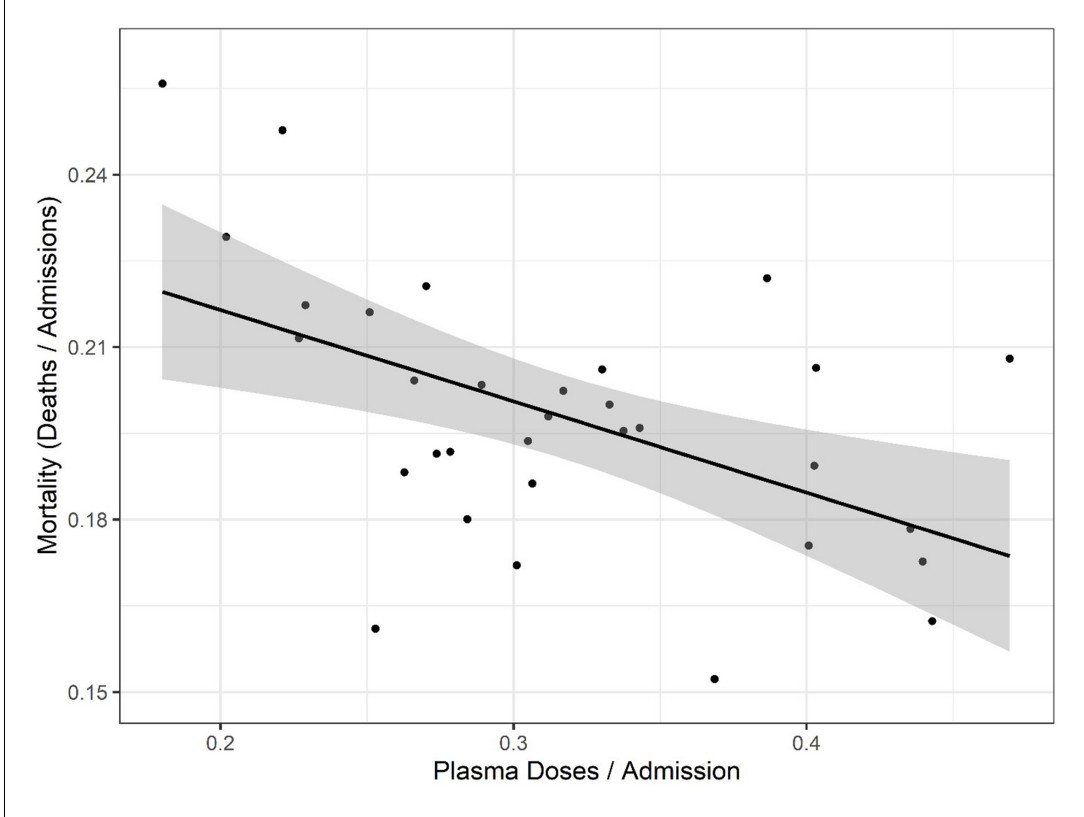

**Figure 3.** Correlation of mortality (death per admission) and COVID-19 convalescent plasma (CCP) doses per admitted patients using the Our World in Data (OWID) database. Correlation analysis yields a Pearson's correlation coefficient of −0.518 (p=0.0024). The black line represents a linear model regression with an R squared of 0.268.

The online version of this article includes the following source data and figure supplement(s) for figure 3:

**Source data 1.** Source data for *Figure 3*.

**Figure supplement 1.** A series of linear regressions and Pearson's correlation tests comparing weekly reported deaths to new weekly hospital admissions, offset by various numbers of weeks to identify the length of lag between admission and death of patients using Our World in Data (OWID) database.

**Figure supplement 1—source data 1.** Source data for *Figure 3—figure supplement 1* .

**Figure supplement 2.** Mortality from COVID-19 by quintile of percent of admissions receiving COVID-19 convalescent plasma (CCP).

**Figure supplement 2—source data 2.** Source data for *Figure 3—figure supplement 2*.

**Figure supplement 3.** Investigation of high age group mortality.

**Figure supplement 3—source data 3.** Source data for *Figure 3—figure supplement 3*.

mortality for COVID-19 patients for the large RECOVERY trial in the UK (*Horby et al., 2021*) and the 30% mortality of patients receiving late CCP in analysis from the EAP (*Joyner et al., 2021*). An extrapolation of the linear model to the situation where every patient is treated with CCP yields a mortality of 7.6%, which is lower than the average mortality in US studies, but still within the range reported (*Supplementary file 1*). However, this extrapolation to maximal use is much less reliable given the absence of points in the y-axis region above an CCP utilization ratio of 0.6 and uncertainty as to whether the relationship is linear in those ranges. Hence, we caution the reader about making any strong inference from this estimate while noting that it is close to the 6.2% mortality reported for COVID-19 patients treated with high titer CCP very early upon hospitalization (*Salazar et al., 2021*). Nevertheless, it is possible to use these efficacy numbers to estimate what the effect on deaths would have been had the USA continued to use CCP at the height of its usage in early Fall 2020, when more than 40% of all patients received plasma therapy.

With this model as a framework for estimating the excess number of deaths, the results of three scenarios were obtained (*Figure 4*). The total observed deaths over the study period was 356,534. Had the rate of CCP utilization observed during August through October 2002 carried over for the

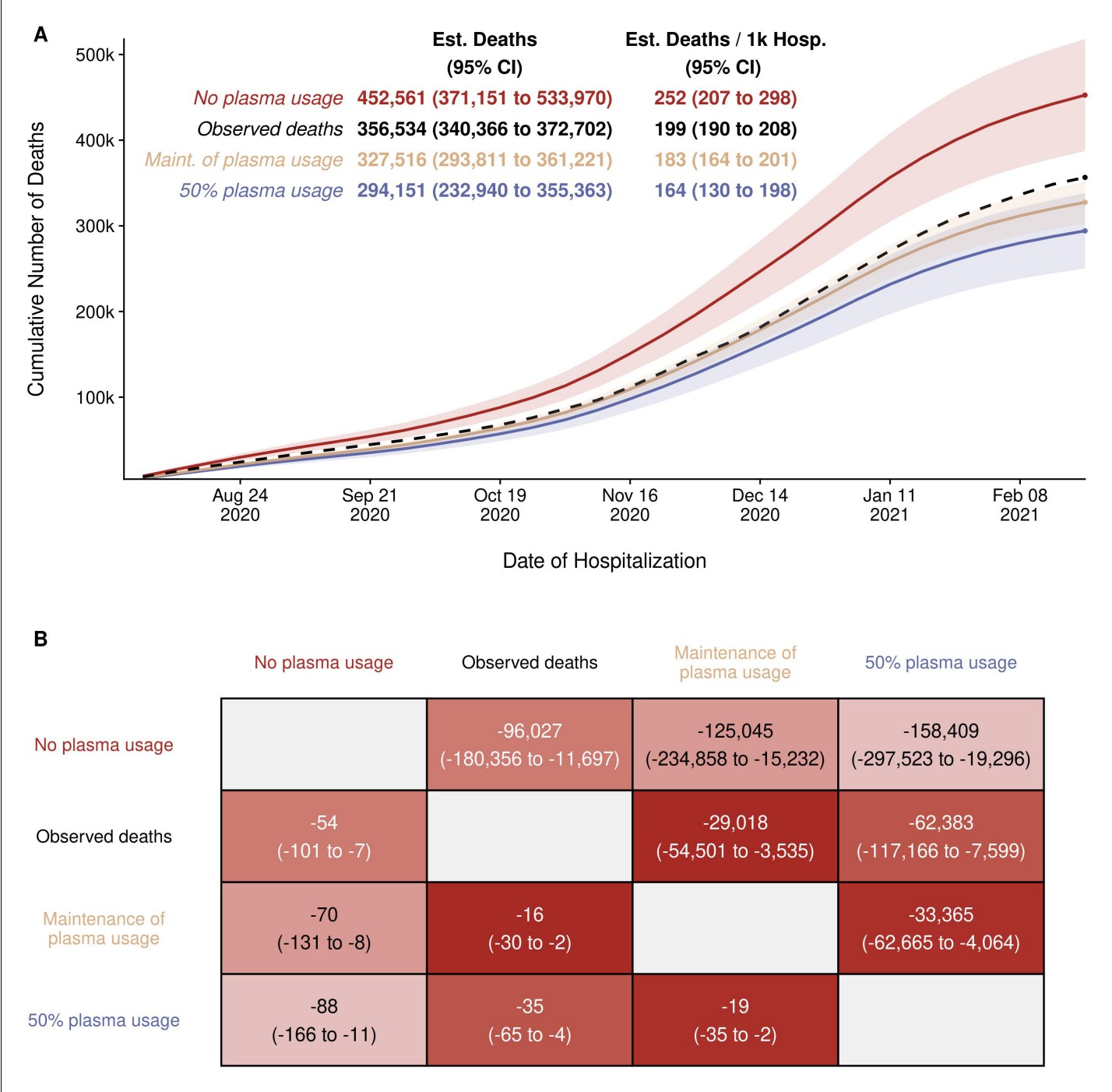

**Figure 4.** Estimated (Est.) deaths under modeled scenarios of COVID-19 convalescent plasma (CCP) using Centers for Disease Control (CDC) database. Panel A presents the longitudinal observed (dashed line) and modeled number of deaths under three scenarios for CCP over the study period (August 3, 2020 to February 22, 2021) that included 356,534 deaths in 1,793,502 hospitalized patients. Over the entire study period, the CCP utilization ratio was 29.1%. In the scenario labeled maintenance (Maint.) of plasma, the CCP utilization ratio was set to 39.5%. With the no plasma and 50% plasma usage scenarios, the CCP utilization ratio was set at 0% and 50%, respectively. Panel B provides the pairwise comparisons of these scenarios to estimate the difference in expected number of deaths among the scenarios for the entire hospitalized patients (upper right triangle) and re-indexed to events per 1000 patients (lower left triangle). The rows represent the comparator or reference scenario, columns indicate the altered CCP use scenario. For example, the cell that intersects the observed deaths and the maintenance of plasma column shows that 29,018 fewer deaths would result had plasma use remained at the 39.5% level.

The online version of this article includes the following source data and figure supplement(s) for figure 4:

**Source data 1.** Source data for *Figure 4*.

*Figure 4 continued on next page*

*Figure 4 continued*

**Source data 2.** Source data for *Figure 4*.
**Source data 3.** Source data for *Figure 4*.
**Figure supplement 1.** Replicated cumulative excess deaths analysis per Our World in Data (OWID) database for scenario 1 (orange).
**Figure supplement 1—source data 1.** Source data for *Figure 4—figure supplement 1*.

remaining months, the expected number of deaths was 327,516 (95% CI: 293,811–361,221), which might result in 29,018 (95% CI: 3535–54,501) fewer deaths than observed. This excess death, in comparison, was small relative to the estimate that results from assuming plasma utilization was as high as it was after the EUA was issued. Had 50% utilization of plasma been continued, 62,383 (95% CI: 7599–117,166) fewer deaths may have been observed. Under the most extreme scenario comparing no plasma use to the highest observed use, a difference of 158,409 (95% CI: 19,296–297,523) deaths is estimated. We repeated this analysis independently for each of the databases used (CDC versus OWID) with concordant results (*Figure 4*; *Figure 4—figure supplement 1*; *Supplementary file 2*; *Supplementary file 3*). The models also estimate that between 94,436 (*Figure 4—figure supplement 1*; *Supplementary file 3*) and 95,026 (*Figure 4*) excess deaths might have occurred in the USA had CP never been deployed using the OWID and CDC databases, respectively.

As an alternative method to estimating the excess deaths, which alleviates the need for a model, the data were summarized into two time periods characterized by high and low utilization of CCP (*Table 1*). In each of the 7 weeks from September 21 to November 8, 2020, the estimated proportion of inpatients transfused with CP exceeded 40% (average 42.6%). We refer to this period as 'high utilization'. In the period immediately following – from November 9, 2021, to the week starting March 22, 2021 – utilization was lower; transfusion rates declined steadily, averaging 27.4% and we refer to this as the 'low utilization' period. The 2-week lagged mortality ratio for 257,424 patients hospitalized in the high utilization period was 18.16% and rose to 20.08% for the 1,344,463 patients admitted in the 18-week period of low utilization. *Table 1* shows that if the mortality rate in the low utilizing 18 weeks had been the same as during the high transfusion period, 25,871 fewer deaths might have taken place.

## Discussion

The use of CCP in hospitalized patients rose steadily after the FDA EUA of late August 2020, and from mid-September to the end of October, we estimate that CCP was provided to more than 40% of admissions. This rise occurred despite opinion pieces arguing that evidence of efficacy was insufficient for the authorization (*Baden et al., 2020*; *Pau et al., 2021*) . However, RCTs from India (*Agarwal et al., 2020*), Argentina (*Simonovich et al., 2021*), and the UK (*Horby et al., 2021*) reporting that CCP use did not reduce mortality in hospitalized patients were followed by a substantial decline in use that began in November. The American Association of Blood Banks reported a 50% increase in the number of institutions planning to stop offering CCP between February and March 2021 (*Anonymous, 2021a*). By March 2021, no more than 10% of hospitalized patients were receiving CCP.

The drop in per capita CCP utilization was strongly associated with an increase in mortality among hospitalized COVID-19 patients. It is notable that mortality from COVID-19 among

**Table 1.** Estimated number of excess deaths due to transfusion hesitancy.

| Time period | Transfusion rate | Number of admissions | Deaths | Mortality rate | Expected deaths if mortality had remained at 18.16% | Excess deaths in the low transfusion period | Excess deaths per 1000 admissions |
|---|---|---|---|---|---|---|---|
| High utilization (September 21, 2021 to November 8, 2021) | 42.59% | 257,424 | 46,747 | 18.16% | 46,747 | – | – |
| Low Utilization (November 9, 2021 to March 22, 2021) | 27.43% | 1,344,463 | 270,019 | 20.08% | 244,148 | 25,871 | 19.2 |

hospitalized patients decreased substantially over most of 2020, consistent with worldwide trends (*Ledford, 2020*), but then began to rise in late November and early December 2020, a period that coincided with reduction in CCP use. If the association we describe is truly causal, we estimate that reduced use of CCP resulted in 29,000–36,000 excess deaths over the past year in the USA.

Concluding that a causal relationship exists between CCP use and mortality requires excluding the contribution of other variables that can affect mortality, a challenging task during an ongoing pandemic when information about the pathophysiology and clinical course of COVID-19 is accruing rapidly. All possible confounders are not available to us to include in this analysis. However, five factors can reasonably be excluded. We excluded that variation in age of hospitalized patients explained the findings since the mean age of the hospitalized population with COVID-19 was actually older during peak use of CCP than later, implying that the association might be even higher if age of admissions was taken into account. Similarly, we considered whether the winter surge in hospital admissions that stretched staff to the utmost could have affected the death rate, but excluded this possibility since the number of hospital admissions per week was not associated with mortality (r = −0.02). A delay in recording of death certificate information could not account for the findings since our findings are robust when deaths are considered from 2 to 5 weeks after admission. We considered whether the emergence of new variants accounted for our findings but excluded this possibility since this concern would apply largely to the most recent months. Only 76 isolations of the UK B.1.1.7 variant, which is associated with higher contagiousness and perhaps, mortality (*Challen et al., 2021*), were identified by January, an estimated 0.5% or less of COVID infections at that time (*Galloway et al., 2021*). Even as late as mid-March 2021, a time beyond the analyses of this paper, the B.1.1.17 variant comprised only 25% of US isolates (*Anonymous, 2021c*); increased mortality from these infections, if present, would not manifest itself until times later than our analysis included. In addition, limiting the end-time of our analysis to January 2021 did not change the findings. Finally, we note that interventions associated with efficacy against COVID-19 such as corticosteroids, Remdesivir, and prone positioning were introduced into clinical practice before the decline in CP use that began in late fall and that no new effective therapy was introduced or withdrawn during the October 2020 to March 2021 period, except for the decline in CP use documented here.

Our findings should be interpreted with caution but several aspects of the evidentiary landscape point to CPP efficacy in reducing mortality when used appropriately. First, an analysis of several dozen studies associated the administration of CCP with reduced mortality, with an overall effect size of ~35% (*Klassen et al., 2021*). These studies include a powerful RCT in elderly outpatients that halved progression to severe disease. In the USA, the two completed RCTs each show a reduction of mortality with CPP usage (*Bennett-Guerrero et al., 2021*; *O'Donnell et al., 2021*). Second, CP has proven effective in individuals with defective humoral immunity and B cell defects (*Senefeld et al., 2020*; *Thompson et al., 2021*). Third, the active agent in plasma, specific antibody to SARS-CoV-2, is strongly related to mortality in transfused patients (*Joyner et al., 2021*; *Libster et al., 2021*; *Maor et al., 2020*). Fourth, the active agent of CCP, specific antibody to SARS-CoV-2, has powerful antiviral activity and has been shown to reduce the viral load in patients with COVID-19, thus providing a mechanistic explanation for its therapeutic effect (*Kim et al., 2020*). Fifth, human CP is protective in murine models of COVID-19 (*Sun et al., 2020*). Sixth, patients treated with CCP manifest reduced inflammatory markers (*Bandopadhyay et al., 2021*; *Tremblay et al., 2020*; *Wu et al., 2021*) that can lead to host damage (*Pirofski and Casadevall, 2020*). Seventh, research has identified nine places where plasma affects the cascade of inflammation to the benefit of treated patients (*Acosta-Ampudia et al., 2021*).

The clinical evidence on CP and mortality is mixed. Three randomized trials have failed to show a reduction in mortality (*Agarwal et al., 2020*; *Horby et al., 2021*; *Simonovich et al., 2021*), another RCT found a 40% reduction in mortality (*O'Donnell et al., 2021*), as other smaller trials and comparative studies (reviewed in *Klassen et al., 2021*). Unfortunately, many trials treated patients with advanced disease. For example, while the overall findings of the RECOVERY trial were null (*Horby et al., 2021*), all four categories of patients treated early in the course of illness (before antibody conversion, in the first 7 days, not receiving steroids, not on oxygen) showed lower mortality in the CPP arm. Late use of plasma, or use in people with advanced illness, is not likely to be effective, because at that time the inflammatory response itself is the major pathophysiologic pathway to severe illness and death, and added antibody is likely powerless to change the course of illness (*Casadevall et al., 2021b*).

While we agree that RCTs are the ideal source of information on treatments, if trials are done in a patient population unlikely to benefit from the intervention because of late disease, they cannot fully contribute to understanding. We argue for taking a more holistic look at the available data on CP, which, as noted above, is consistent with benefit of treatment in early disease and with adequate antibody in the plasma. Our population-level analysis is analogous to the observational studies that linked smoking and lung cancer, a finding universally accepted by the weight of evidence from many sources, in spite of the absence of any RCT demonstrating the finding (*Smoking and Health, 1964*).

Taking a page from the reluctance of citizens to accept vaccines against SARS-CoV-2, a phenomenon that has been termed 'vaccine hesitancy' (*Pullan and Dey, 2021*), we call the reduced use of CCP 'plasma hesitancy'. Plasma hesitancy may be a result of health care providers overvaluing and overemphasizing negative results from RCT findings while dismissing other evidence that CCP reduces mortality. Physicians have relatively little recent experience with treating infectious diseases with antibody therapies (*Casadevall et al., 2021b*). Chemical agents, however, are more familiar, and the success of antiviral drugs against human immunodeficiency and hepatitis C viruses is well known.

CCP remains one of few options available for treating COVID-19 early in the disease. It is readily available wherever there are recovered patients, requires no complex manufacturing, is inexpensive, and has an excellent safety profile (*Joyner et al., 2020a*). Interim guidelines for the American Association of Blood Banks (*Cohn et al., 2021*) and from Brazil (*De Santis et al., 2021*) emphasize both of these principles – early treatment and high antibody content in plasma. We hope that physicians policymakers, regulators, and guideline committees consider the totality of the available evidence, including our findings, when making decisions for CCP use in individual patients (*Casadevall et al., 2021a*).

## Acknowledgements

We are grateful to Bill Block and Jennifer Kapral from BCA (West Warwick, RI) and Kate Fry from America Blood Centers (Washington, DC) for their help in obtaining the plasma usage data. Disclaimer: The views and opinions expressed in this publication are those of the authors and do not reflect the official policy or position of the US Department of Health and Human services and its agencies including the Biomedical Research and Development Authority and the Food and Drug Administration, as well as any agency of the US government. Assumptions made within and interpretations from the analysis are not reflective of the position of any US government entity.

## Additional information

### Funding

| Funder | Grant reference number | Author |
| --- | --- | --- |
| National Institute of Allergy and Infectious Diseases | AI1520789 | Arturo Casadevall |
| Biomedical Advanced Research and Development Authority | 75A50120C00096 | Michael J Joyner |

The funders had no role in study design, data collection and interpretation, or the decision to submit the work for publication.

### Author contributions

Arturo Casadevall, Conceptualization, Data curation, Formal analysis, Investigation, Methodology, Writing - original draft, Project administration; Quigly Dragotakes, Formal analysis, Investigation, Methodology, Writing - original draft; Patrick W Johnson, Formal analysis, Investigation, Methodology; Jonathon W Senefeld, Stephen A Klassen, Investigation, Writing - review and editing; R Scott Wright, Conceptualization, Project administration, Writing - review and editing; Michael J Joyner, Conceptualization, Investigation, Writing - original draft, Project administration; Nigel Paneth, Conceptualization, Formal analysis, Investigation, Methodology, Writing - original draft; Rickey E Carter,

Conceptualization, Data curation, Formal analysis, Investigation, Methodology, Writing - original draft

### Author ORCIDs
Arturo Casadevall (iD) https://orcid.org/0000-0002-9402-9167

### Decision letter and Author response
Decision letter https://doi.org/10.7554/eLife.69866.sa1
Author response https://doi.org/10.7554/eLife.69866.sa2

## Additional files

### Supplementary files
• Source data 1. Centers for Disease Control (CDC) analysis data. Summarized data of hospitalizations and deaths, from CDC data.

• Source data 2. Blood banker distribution data for plasma units.

• Supplementary file 1. Mortality of COVID-19 in US COVID-19 convalescent plasma (CCP) efficacy studies.

• Supplementary file 2. Data from and calculations for excess mortality from COVID-19 convalescent plasma (CCP) hesitancy based on the Centers for Disease Control (CDC) database.

• Supplementary file 3. Excess death calculations based on Our World in Data (OWID) database.

• Transparent reporting form

### Data availability
All data generated or analysed during this study are included in the manuscript and supporting files.

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
