## [Decision Letter]

**Acceptance summary:**

This work is of interest to clinicians, epidemiologists and policy makers as it raises concerns about under-utilization of convalescent plasma (CCP) therapy during the Covid-19 pandemic, which in turn led to an increased number of preventable patient deaths. The authors demonstrate an inverse correlation between CCP use and mortality per admission in the US. They estimate that reduced use of CCP may have resulted in 29,000 to 36,000 excess deaths over the past year in the US.

**Decision letter after peer review:**

Thank you for submitting your article "Convalescent Plasma Use in the United States was inversely correlated with COVID-19 Mortality" for consideration by *eLife*. Your article has been reviewed by 3 peer reviewers, one of whom is a member of our Board of Reviewing Editors, and the evaluation has been overseen Mone Zaidi as the Senior Editor. The reviewers have opted to remain anonymous.

The Reviewing Editor has drafted this to help you prepare a minor revised submission.

Essential revisions:

Overall all three reviewers were enthusiastic for the manuscript and largely felt that it is appropriate for publication. However, two of the reviewers had very minor comments and suggestions, which should be able to be easily corrected and a revised document should be acceptable without further review.

*Reviewer #2 (Recommendations for the authors):*

This paper provides an important analysis of the correlation between CCP use and patient deaths from Covid-19 and has significant implications for future use of this therapeutic option. The authors are urged to consider the following:

1) The manuscript clearly makes the distinction between correlation and causation in most places; however, the authors should carefully review the manuscript and ensure that this is done consistently. For example, on line 182, please consider "…use these 'implied' efficacy numbers…'might' (rather than would) have been…". Similarly, on line 203, please consider replacing "would have taken place" with "might have taken place" or "would be predicted by the model".

2) The statement on line 279 urges physicians to consider these findings when making clinical use decisions. Do the authors also urge policymakers, regulators, and guideline advisory groups to do so as well?

3) The analysis of cofounding variables (starting on line 219) is good, but of course there are other considerations as well. Could you make a comment about the potential impact of the changing use of other therapies (remdesivir, steroids, etc) or clinical interventions (prone position, etc)?

4) Is the graded mortality shown in supplemental figure 3 statistically significant?

*Reviewer #3 (Recommendations for the authors):*

The definitions of "high utilization" and "low utilization" need to be explained in the text for Table 1, as those specific terminologies are not defined in the table.

The text in figure 4 is difficult to read, and recommend re-working the figure to make comprehension easier.

---

## [Author Response]

Reviewer #2 (Recommendations for the authors):This paper provides an important analysis of the correlation between CCP use and patient deaths from Covid-19 and has significant implications for future use of this therapeutic option. The authors are urged to consider the following:1) The manuscript clearly makes the distinction between correlation and causation in most places; however, the authors should carefully review the manuscript and ensure that this is done consistently. For example, on line 182, please consider "….use these 'implied' efficacy numbers…'might' (rather than would) have been…". Similarly, on line 203, please consider replacing "would have taken place" with "might have taken place" or "would be predicted by the model".

Changes to the text made as recommended by the reviewer. Also, carefully reviewed the text to avoid any hint that the data implied causation.

2) The statement on line 279 urges physicians to consider these findings when making clinical use decisions. Do the authors also urge policymakers, regulators, and guideline advisory groups to do so as well?

Agree. Sentence was modified to specifically mention policymakers, regulators, and guideline advisory groups (line 307).

3) The analysis of cofounding variables (starting on line 219) is good, but of course there are other considerations as well. Could you make a comment about the potential impact of the changing use of other therapies (remdesivir, steroids, etc) or clinical interventions (prone position, etc)?

Agree with suggestion and a new sentence was added (lines 262-266).

“Finally, we note that interventions associated with efficacy against COVID-19 such as corticosteroids, Remdesivir and prone positioning were introduced into clinical practice before the decline in CP use that began in late fall and that no new effective therapy was introduced or withdrawn during the October 2020 to March 2021 period, except for reduced CP use documented here.”

4) Is the graded mortality shown in supplemental figure 3 statistically significant?

Yes. R and p values added to figure and figure legend.

Reviewer #3 (Recommendations for the authors):The definitions of "high utilization" and "low utilization" need to be explained in the text for Table 1, as those specific terminologies are not defined in the table.

Agree. We have amended the text to define high and low utilization. (lines 218 to 222)

The text in figure 4 is difficult to read, and recommend re-working the figure to make comprehension easier.

Figure 4 was revised with larger font. In addition, we revised Supplementary Figure 5 to increase its font size since it was also too small.